# MAKE IT COUNT: TEXT-TO-IMAGE GENERATION WITH AN ACCURATE NUMBER OF OBJECTS

## ABSTRACT

Despite the unprecedented success of text-to-image diffusion models, controlling the number of depicted objects using text is surprisingly hard. This is important for various applications from technical documents, to children's books to illustrating cooking recipes. Generating object-correct counts is fundamentally challenging because the generative model needs to keep a sense of separate identity for every instance of the object, even if several objects look identical or overlap, and then carry out a global computation implicitly during generation. It is still unknown if such representations exist. To address count-correct generation, we first identify features within the diffusion model that can carry the object identity information. We then use them to separate and count instances of objects during the denoising process and detect over-generation and under-generation. We fix the latter by training a model that predicts both the shape and location of a missing object, based on the layout of existing ones, and show how it can be used to guide denoising with correct object count. Our approach, *CountGen*, does not depend on external source to determine object layout, but rather uses the prior from the diffusion model itself, creating prompt-dependent and seed-dependent layouts. Evaluated on two benchmark datasets, we find that CountGen strongly outperforms the count-accuracy of existing baselines.

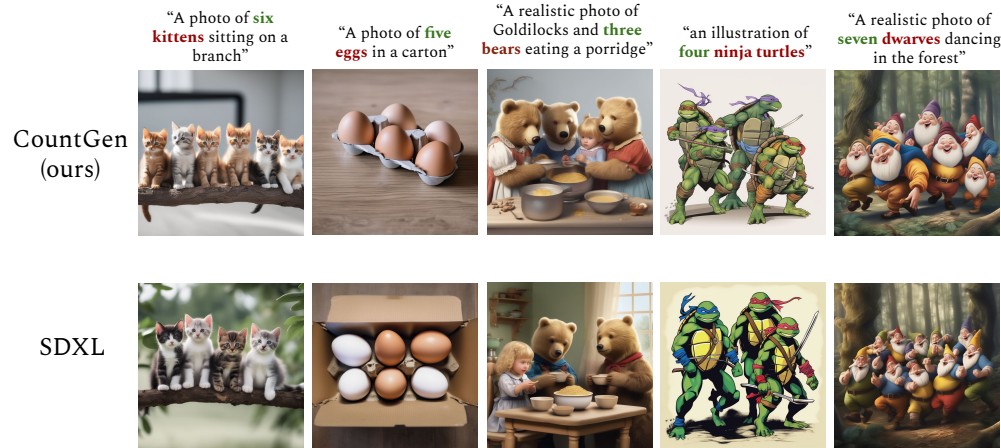

Figure 1: **CountGen** generates the correct number of objects specified in the input prompt while maintaining a natural layout that aligns with the prompt.

## 1 INTRODUCTION

Text-to-image diffusion models provide an accessible way to control the generation of visual content. A major failure mode is their inability to count, that is, they often fail to generate the correct number of items in response to text prompts. For instance, when asked to generate an image of Goldilocks and the three bears, models may generate only two bears (Figure 1). Counting failures are particularly frustrating: The accuracy is surprisingly low, and mistakes are often obvious for people to detect.

To illustrate the difficulty of the problem, consider some naive attempts to work around it. First, one can manually design layouts per count, to determine the spatial organization and the number of objects, then provide it as a conditioning signal to a generative model (Dahary et al., 2024b). This approach would fail to generate prompt-dependent layouts, which is highly desirable. One could also try asking large vision-language models to propose layouts (e.g., Chen et al. (2023); Feng et al. (2023)), but these approaches do not use the visual prior information that text-to-image models already collected, and, as we show below, their performance is rather poor for the counting task.

Why is it so hard for diffusion models to count while they generate? First, counting objects requires that models capture "objectness" – the high-level coherent representation of something being a separate entity, even if surrounded by other similar entities. Capturing objectness is by itself a hard task in image understanding (Alexe et al., 2012; Kuo et al., 2015), and long studied in cognitive psychology (Spelke, 1990). It is currently not known to what extent diffusion models represent objectness of entities they generate. A second main challenge is that text-to-image diffusion models struggle with controlling spatial layout just from text. Producing a correct number of objects requires obeying a global and complex spatial relation between entities in an image (Chefer et al., 2023; Dahary et al., 2024a).

To address the problem of accurate count generation we describe several new contributions, which together form our method *CountGen*. First, we analyze the representations of the self-attention layers in SDXL (Podell et al., 2023), and identify features that capture objectness and instance identity. We then use these features to develop ways to detect instances of objects during the denoising process, find their spatial layout and count them. Specifically, we localize the features that correspond to objects using the cross-attention maps and cluster them to form object instance segmentation. Importantly, we do not have to wait for an image to be fully generated, and we can accurately count the number of objects already at an intermediate step of the denoising process.

Given this new capability to count the number of objects being generated during the denoising process, we further develop methods to correct generation when the count does not match the prompt. First, we train a layout-modification network we call ReLayout. It takes a spatial layout of $k$ objects and generates a similar spatial layout with one more instance of an object added in a natural location for the input layout. For example, given a row of five kittens sitting on a branch, it learned to add a sixth kitten to the same row. This model is trained using image-pair samples generated by the diffusion model itself. Finally, we show how to use the new layouts in a new test-time-optimization procedure.

We evaluate CountGen on text prompts from the T2I-CompBench (Huang et al., 2023) which includes prompts with numbers. CountGen greatly improves accuracy, as evident by human evaluation experiments, from 29% accuracy for SDXL to 48% by our method. It also improves over all other baseline methods including large commercial models like the recent DALL-E 3 (Betker et al., 2023). To support future work in this field, we design and release a dataset that can be evaluated automatically. Specifically, we release CoCoCount, a set of prompts based on COCO classes(Lin et al., 2014), which can easily be evaluated using COCO-trained object detectors, like YOLO (Wang et al., 2024). CountGen also significantly improves over all baseline methods on CoCoCount, importantly from 26% accuracy for SDXL to 52% by our method.

In summary, this paper makes the following new contributions (1) We identify novel features that represent objectness and instance identity in SDXL (Podell et al., 2023). (2) We design an inference-time optimization to guide SDXL to generate an accurate number of instances for an object. (3) We describe a learning approach to automatically modify layouts to add a new instance of an object while preserving the structure of the scene. (4) We achieve state-of-the-art results in count-accurate generation.

## 2 RELATED WORK

**Generating images with accurate object count.** Numerous papers noted that text-to-image diffusion models often fail to produce images that accurately match text prompts, especially when these prompts specify an exact number of objects (Kang et al., 2023; Zhang et al., 2023; Paiss et al., 2023; Wen et al., 2024; Battash et al., 2024; Feng et al., 2023; Lee et al., 2023; Fan et al., 2023; Sun et al., 2023; Rassin et al., 2022; Dahary et al., 2024a; Rassin et al., 2024; Chefer et al., 2023). Various efforts were made to improve the accuracy of these models. Most relevant to our work, Kang et al. (2023)

proposed a classifier-guidance approach to improve object count accuracy. The method "counts" instances at each diffusion step using a pretrained counting network and adjusts the denoising process using gradient guidance. However, it requires using an additional U-Net in *every* denoising step.

An important line of work suggests breaking the generation process into two steps: (1) Text-to-layout - setting a spatial location for every object instance; and (2) Layout-to-image - generating an image with the correct object count using the given layout. **Text-to-Layout:** Several studies used large language models (LLMs) to propose spatial layouts (Chen et al., 2023; Phung et al., 2023; Feng et al., 2023; Gani et al., 2024). LayoutGPT (Feng et al., 2023) injects visual commonsense into the LLM prompt which enables it to generate desirable layouts. Gani et al. (2024) suggest decomposing complex prompts into smaller prompts before injecting them into the LLM. **Layout-to-image:** Providing a predefined layout with the exact number of subjects helps ensure that the generated images reflect the intended count (Chen et al., 2023; Yang et al., 2023). Bounded Attention (Dahary et al., 2024a) addresses this challenge by channeling attention to bounding boxes corresponding to object instances. However, this approach requires users to manually provide the bounding boxes for all the instances of each object. In contrast to these separate-step approaches, CountGen, addresses the two steps of count-accurate generation. It first corrects the layout that emerges during generation so it contains the correct number of instances. It then uses a novel test-time optimization method to generate a count-accurate image.

**Controlling text-to-image models through attention-based loss.** To address the issue of object neglect—when objects mentioned in a text prompt fail to appear in the generated image— Chefer et al. (2023) developed a novel loss function that ensures all objects in the prompt are reflected in the cross-attention maps used during image generation. Rassin et al. (2024) tackled the challenge of incorrect attribute association by designing a loss function that binds the cross-attention maps of subjects and their attributes more effectively. Inspired by these advancements, CountGen includes a novel cross-attention maps loss function designed to ensure the generation adheres closely to the input layout.

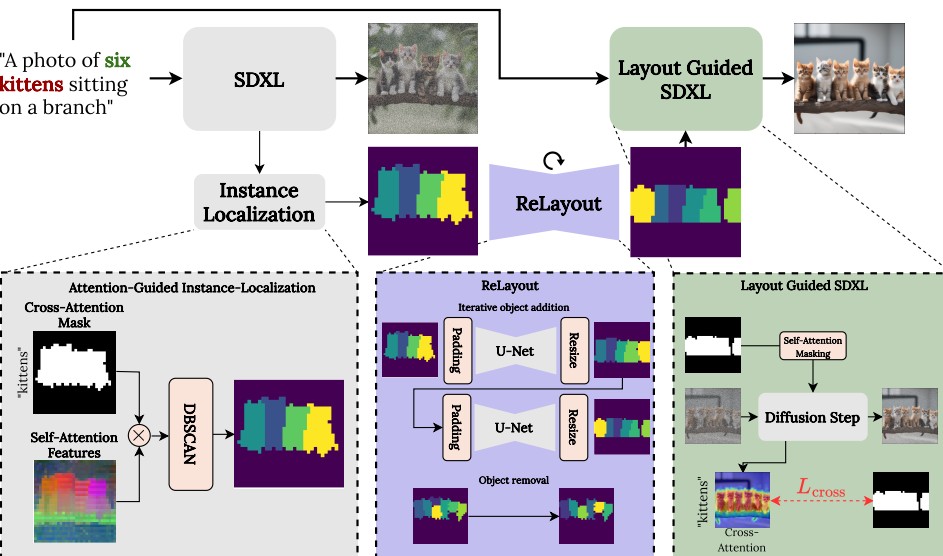

Figure 2: **Architecture outline:** Given a prompt that includes a quantity, we begin generating a corresponding image using pretrained SDXL until timestep $t = 500$. We then perform **Instance Localization**, where we combine cross-attention maps corresponding with the object, and self-attention features extracted at timestep $t$ to generate object clusters for each generated object. Then we apply **ReLayout**, which generates an object layout with the correct number of instances, while preserving the composition of the extracted layout. Finally, we perform **Layout Guided** generation, which applies an inference time optimization based on the layout through cross-attention loss $L_{\text{cross}}$ and self-attention masking.

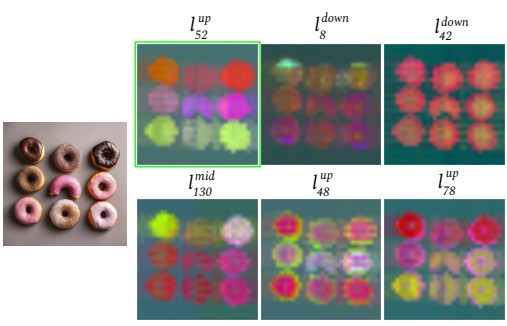

Figure 3: **PCA visualization.** to explore the notion of objectness inside SDXL latent space, we visualize dimension-reduced self-attention feature maps from various layers across the network at timestep $t = 500$. We notice that although most layers do not exhibit a clear separation of objects, layer $l_{52}^{up}$ displays a robust separation indicated by different object instances having distinct colors. Visualization across different timesteps is shown in the appendix, Figure 10.

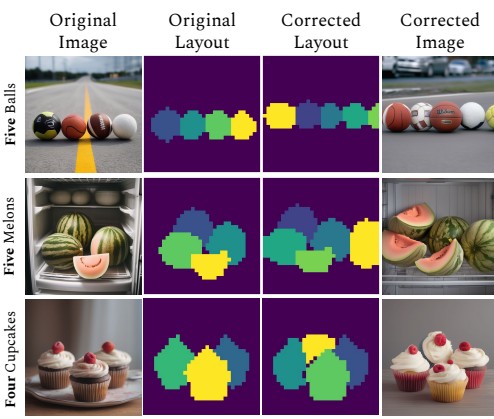

Figure 4: **Correcting under-generation.** we show examples for the ReLayout correction of cases where SDXL generates less objects than specified in the prompt. It is evident that the generated layouts are natural and obey the same composition of the original generation, with the correct number of objects.

## 3 OUR APPROACH: COUNTGEN

Our method, CountGen, aims to enhance text-conditioned image generators to accurately produce the intended number of objects for complex input prompts. Our methodology involves a two-step process: initially, we generate a *natural* layout that specifies where and how objects should appear in the image Section 3.2. That layout is based on a layout that emerges naturally from the text-conditioned generation (Section 3.1). At the second step (Section 3.3), we use this layout as a blueprint to generate the final image.

### 3.1 DISCOVER OBJECT-INSTANCE LAYOUT DURING EARLY GENERATION

To count object instances during generation, one must first find an internal representation that captures the separate identity of different object instances. It is not known if this representation exists in diffusion models like SDXL. We now discuss this representation and then show how we can detect the layout of object instances during early generation.

**An emerging instance-identity representation in SDXL.** We begin by exploring the notion of 'objectness' in SDXL. While previous work (Chefer et al., 2023; Hertz et al., 2023; Tewel et al., 2023) utilized the cross-attention mechanism to localize *objects of a given class* in generated images, little research has been conducted on whether the model encodes information about *object instances* and how to distinguish between different instances of an object. We tackle this problem by exploring a variety of features across different layers and timesteps of the diffusion process to determine if and where the model encodes instance-level information. Figure 3 illustrates this analysis using PCA visualization of self-attention features from various layers across SDXL at timestep $t = 500$, which shows the most robust instance representation (See appendix Figure 10). While most layers do not exhibit separability at the instance level, we notice that layer $l_{52}^{up}$ tends to generate different features for different instances of the same object, with each instance having its distinct color. Based on this finding, we select the self-attention features from layer $l_{52}^{up}$ at timestep $t = 500$ to serve as our instance-level features.

**Identifying object instances.** Building on the findings of Hertz et al. (2023), which show that cross-attention maps can pinpoint a token's position in a generated image, we create a *foreground mask* for each object described in the prompt. By contrasting these foreground masks, derived from the cross-attention, with the self-attention features, we effectively segregate pixels associated with objects from those belonging to the background. Subsequently, we cluster the object-associated

pixels from the self-attention map into distinct masks for each object. This approach allows us to refine our object representations and enhance the accuracy of the generated layouts.

Formally, let $A_{l,t}^{self}, A_{l,t}^{cross}$ represent the self-attention and cross-attention maps, respectively, for layer $l$ at timestep $t$ within our diffusion network. We aggregate cross-attention maps associated with the tokens corresponding to the objects specified in the input prompt. We then use these cross-attention maps to extract a foreground mask $M$ based on dynamic thresholding $M = \text{Otsu}(A_{l,t}^{cross})$, where Otsu applies the Otsu thresholding method (Otsu, 1979; Tewel et al., 2024) to determine foreground (object) pixels. We define $p_k \subseteq A_{l,t}^{self}$ as the set of features from the self-attention map that are identified as foreground by mask $M$. We then cluster these patches: $Clusters = DBSCAN(p_k, \epsilon)$, where $DBSCAN(\cdot, \epsilon)$ is the DBSCAN (Ester et al., 1996) clustering algorithm with a dynamic parameter $\epsilon$. Finally, the initial layout $L$ is created by grouping the object clusters: $L = \bigcup_{C \in Clusters} C$.

At the end of this process, we obtain a set of masks, one for each object being generated. This is illustrated in Figure 2, left gray box.

### 3.2 RELAYOUT: CORRECTING THE NUMBER OF OBJECTS IN THE MASK

We now introduce our layout-correction component, *ReLayout*, which **preserves the overall scene composition** while correcting the number of objects. For example, Figure 2 depicts an image generated using the prompt "a photo of six cats", but only four cats were generated. Our ReLayout generates a new layout with the correct number of instances while keeping the overall composition of the kittens sitting in a row. More examples are shown in Figure 4.

The input to the ReLayout is an object-layout described in Section 3.1, from which we initially infer the number of generated instances. Next, our ReLayout component takes one of two corrective actions based on the discrepancy between the generated and expected counts. In cases of **over-generation**, where more instances were generated than requested, ReLayout deterministically removes the smallest instances to achieve the desired cluster count. We find that this simple strategy produces appealing results. In cases of **under-generation**, a more intricate challenge arises: the ReLayout must insert new instances to the scene in a way that preserves the original scene structure. This process involves a sophisticated understanding of different object layouts—like the stark contrast between linearly arrayed *bottles* and the clustered arrangement of *elephants*—to seamlessly augment the layout. In Section 3.2.1, we detail our approach for handling under-generation. In cases where the number of instances is correct, the ReLayout maintains the initial layout.

### 3.2.1 HANDLING UNDER-GENERATION

To address under-generation issues, we train a U-Net model to predict a new layout, represented as a multi-channel mask, from an existing layout. In practice, each forward pass of the U-Net generates a mask with an additional instance. This process is applied in iterations until the mask reflects the correct number of instances. In what follows we provide detailed information on the architecture and training of our U-Net model.

**Creating a training dataset.**  To train our ReLayout U-Net, we need a dataset of layout pairs with $k$ and $k+1$ objects, that maintain the same scene composition. We begin with the empirical observation that slight variations in the object count specified in the prompt—while keeping the starting noise and the rest of the prompt consistent—typically results in images with similar layouts, as shown in appendix Figure 9. This consistency is crucial as it allows us to generate a training dataset of layout pairs where each pair has a similar object composition, differing by only one object, thereby preserving the overall scene structure.

Following this observation, we generate a set of ~10K pairs of images of $I_k$ and $I_{k+1}$, where each pair consists of images that differ by only one in the number of objects depicted. Each pair is generated with random fixed seeds and prompts that fit the same template, such as "a photo of two cats" versus "a photo of three cats". To confirm that each image pair accurately represents an $k$ and $k+1$ object scenario, we extract object masks $M_k$ and $M_{k+1}$ as described in Section 3.1, and verify the object count in one image is exactly one more than in its paired image. Overall, the final dataset for training consists of pairs of binary masks $(M_k, M_{k+1})$, representing the U-Net task of learning to generate a mask with $k+1$ objects from a mask with $k$ objects.

**Matching objects.** To train the U-Net, we need to establish a correspondence between each object $i$ in $M_k$ to its new position in $M_{k+1}$. We aim to find a matching that minimizes the shift in objects positions. We use the Hungarian algorithm (Kuhn, 1955) to find the optimal matching. More details in Appendix C.1.

**Training the U-Net module.** We trained the U-Net architecture by adapting it to handle 9 input channels – corresponding to the source tensor $M_k \in \{0,1\}^{W \times H \times k}$ with $k$ objects, and output 10 channels – for the target tensor with $k+1$ objects, to support counts up to 10. We optimized the U-Net parameters using two loss functions: (1) A Dice loss (Sudre et al., 2017) between a predicted masks $\hat{M}_{k+1}$ and the target masks $M_{k+1}$ of that object; and (2) Mask-to-mask overlap loss, designed to reduce the overlap between output masks of different instances. Specifically, this was computed as $1 - L_{Dice}$ between all pairs of predicted masks $\hat{M}^i{}_{k+1}, \hat{M}^j_{k+1}$.

$$\mathcal{L} = \mathcal{L}_{\text{DICE}} + \lambda \mathcal{L}_{\text{overlap}} \tag{1}$$

with $\lambda$ being a weighing hyperparameter. Detailed definitions are provided in Appendix C.2, and evaluation of ReLayout is in Appendix C.3.

**Inference.** At inference time, as a pre-processing step, we first add padding to input masks. After each iteration, we gradually and consistently increase the padding size around the original masks. This operation is beneficial when we need to add a large number of objects, as it creates a "zoom-out" effect, making space for new objects.

We also slightly erode instance masks after the ReLayout module is applied, to improve separation of contacting objects.

## 3.3 CountGen Image: Layout-based Image Generation

Provided with correct object mask layouts (Section 3.2), our goal is to guide the image generation process to adhere to the input layout. Given a mask for each object in the desired layout, we apply an inference time optimization to match the layout in the generated image. To optimize object layouts at inference time, we propose a dual approach: object layout loss to encourage object creation in the foreground, i.e. pixels within the object masks, and self-attention masking to prevent object generation in the background.

**Object layout loss.** Consider the optimization of object placement within layouts using a weighted binary cross-entropy loss. Given $c$, the aggregated cross-attention scores, and $m$, a binary mask denoting object presence (foreground), the weighted binary cross-entropy loss is computed pixel-wise and is defined as follows:

$$L(c, m) = -\sum_i w_i \left( m_i \log c_i + (1 - m_i) \log(1 - c_i) \right),$$

where $c_i$ is the cross-attention score at pixel $i$, $m_i$ is the value of the binary mask at pixel $i$, and $w_i$ is the weight assigned to each pixel $i$ where $w_i = 10$ if $m_i = 1$, otherwise $w_i = 1$. During the SDXL generation process, each step takes a noised latent $X_t$ as input. For the first 25 generation steps, we propagate gradients from the object layout loss to $X_t$, updating it to reduce the loss.

**Self-attention masking.** The object-layout loss encourages objects to be generated in the foreground, but when applied on itself, generated objects may appear outside the object masks (Figure 6). To address this, we mask the self-attention connections between pixels in the background to pixels in the foreground. By disrupting these links, we stop the information flow from the objects to the rest of the image and prevent the model from forming objects in the background. Formally, at layer $l$ and timestep $t$, the masked self-attention $S_t^{*(l)}$ is defined as:

$$S_t^{*(l)}[i,j] = \begin{cases} 0 & \text{if } i \in \mathcal{B}^{(l)} \text{ and } j \in \mathcal{F}^{(l)}, \\ S_t^{(l)}[i,j] & \text{otherwise.} \end{cases}$$

where $i$ and $j$ are pixels indices, $\mathcal{B}^{(l)}$ and $\mathcal{F}^{(l)}$ represents the set of pixels belonging to the background and the foreground respectively, and $S_t^{(l)}$ is the self-attention map at layer $l$ and timestep $t$. We discuss implementation details and computational efficiency in Appendix A.

## 4 EXPERIMENTS

**Compared methods.** We compare CountGen against seven baseline methods: **(1) SDXL** (Podell et al., 2023); **(2) Repeated Object:** SDXL, with a modified prompt, where an object is repeated in the prompt the number of times it is required to generate, as in replacing "three cats" with "a cat and a cat and a cat". This is a naive approach that parallels prompts like "A cat and a dog". **(3) Reason Out Your Layout:** (Chen et al., 2023) uses GPT-3.5 (Brown et al., 2020) to generate layouts then trained an adapter to integrate it to SD-1.4 (Rombach et al., 2022); **(4) DALL-E 3** (Betker et al., 2023); **(5) Random masks + BoundedAttn :** generate a layout with the correct amount of clusters placed randomly in the image and apply a layout-guidance generation method on top; **(6) Counting Guidance (Kang et al., 2023) :** boost generation of SD with a counting network; **(7) RPG (Yang et al., 2024):** generates the layout using GPT-4 and then uses SDXL.

Full details on how we used these baselines are given in Appendix B.3. We also compared our layout-to-image phase, CountGen-Image, described in section Section 3.3 with Bounded Attention (Dahary et al., 2024a).

**Datasets.** We evaluate our method and the baselines using two datasets. **(1) T2I-Compbench-Count.** A subset of T2I-Compbench (Huang et al., 2023), which is a benchmark for open-world compositional text-to-image generation. This subset specifically includes 218 prompts that specify a single object and its number (between 2 to 10). **(2) CoCoCount (ours).** We generate a dataset with automatic evaluation in mind. Specifically, we sample classes from COCO, which are more favorable to accurate and automatic detection by methods, like YOLOv9 (Wang et al., 2024). We design simple prompts around these classes, with a number between 2 and 10. In total, there are 200 prompts with various classes, numbers and scenes. See full details in Appendix C.4.

**Count accuracy evaluation.** We evaluate the results of CountGen and the baselines using human and automatic evaluation method, which is standardized and reproducible. In both settings, we seek to identify if the number of instances generated by the object matches the request in the prompt.

*Human evaluation.* We quantified the count-accuracy of our method and baselines using human raters. Raters were asked for every image: (1) Is the object in the image?; (2) Are its instances well-formed?; (3) How many instances of the object are in the image? If the answer to question (1) or (2) is "no", then we do not ask question (3). We provide details on the platform, rater selection and pay, and screenshots of the task in Appendix D.1.

*Automatic evaluation.* For automatic evaluation, we use the YOLOv9 model (Wang et al., 2024) with its default settings, as it represents the current state-of-the-art in the YOLO object detection benchmarks. To extract the number of objects in the image, we simply count the number of detected bounding-boxes corresponding to the target object.

**Image quality evaluation.** Forcing the diffusion model to obey the count in the text prompt is inevitably expected to reduce the naturalness and visual appeal of generated images, simply because more constraints are added. This effect has been observed in other studies using test-time optimization (Rassin et al., 2024; Chefer et al., 2023). We evaluate the image quality of CountGen by presenting human raters with two images, by CountGen and SDXL, and asking them to select whether one image is more natural and well-formed than the other or to indicate that both images are equally good.

## 5 RESULTS

**Quantitative results.** Table 1 compares CountGen with competing baselines, showing its significant improvement over baselines in both CoCoCount and T2I-compbench-Count. Figure 7, and Figure 12 show CountGen outperforms all baselines for all values, except for two and three instances, where DALL-E 3 slightly outperforms. We hypothesize that DALL-E 3 is larger and was trained on higher-quality data than SDXL (our base model). In terms of image quality, out of 200 comparisons, in only 23 cases the majority of the raters preferred SDXL over our model. This indicates there is no significant loss of quality. We also include the confusion matrix figure of CountGen based on human evaluation in Figure 17.

Table 1: Generated count accuracy. Values are the percent of generated images that have the correct number of objects, for CoCoCount and T2I-Compbench-Count.

| Model | CoCoCount | | T2I-Compbench |
| | YOLOv9 Accuracy | Human Accuracy | Human Accuracy |
|---|---|---|---|
| SDXL | 28 | 26 | 29 |
| Repeated Object | 17 | 18 | 14 |
| Reason Out Your Layout | 21 | 26 | 15 |
| DALL-E 3 | 25 | 38 | 36 |
| Random masks + BoundedAttn | 29 | 30 | 35 |
| Counting Guidance | 21 | 22 | 22 |
| RPG | 21 | 28 | 25 |
| CountGen (ours) | **50** | **52** | **48** |

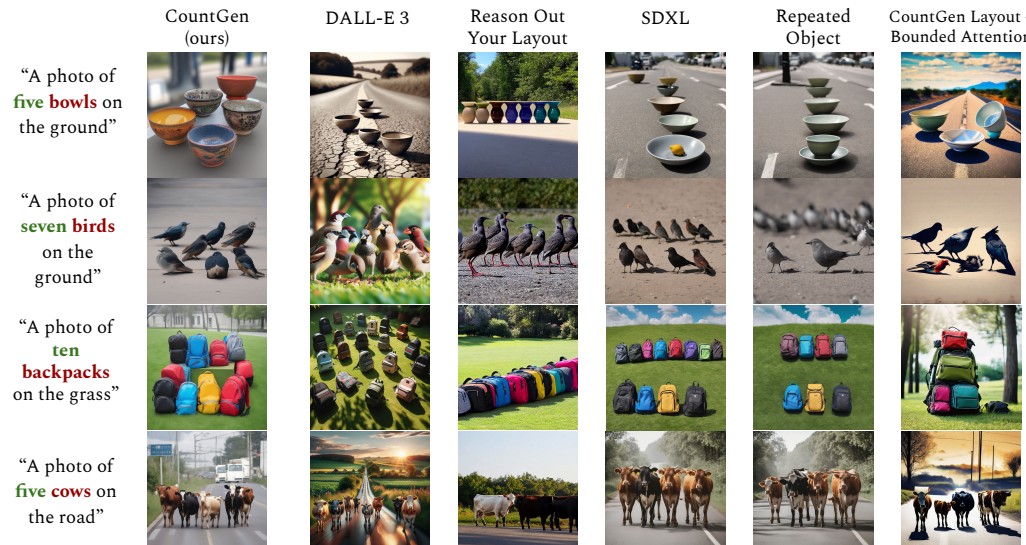

Figure 5: **Qualitative comparisons.** We evaluated CountGen against DALLE 3, Reason Out Your Layout, SDXL, Repeated Object SDXL and Counten Layout + Bounded Attention. Our method successfully generates the correct number of objects, while other methods struggle in some or all of the examples. Additional results are shown in the supplemental material.

**Qualitative results.** Figure 5 shows examples of prompts and the images generated by various methods. In contrast to other methods, CountGen consistently generates the correct object number.

## 6 ABLATION STUDY

**Contribution of *CountGen-Layout* and *CountGen-Image*.** Table 2 quantifies the contribution of each of these components to the overall accuracy, by replacing it with a baseline alternative. Compared with a baseline (Random Masks + Bounded Attention) our first phase CountGen-Layout improves accuracy measured by people by 14% (from 30 to 44), and our second phase CountGen-Image by 12%. Together, the two components add up to improve accuracy by 21 points.

**Layout guided generation ablation study.** The second phase of our method, CountGen-Image, consists of two components: self-attention masking and object layout loss, as described at Section 3.3. To evaluate the contribution of each component, we deactivate it and compare the results. In Figure 6, we qualitatively observe that removing the layout loss leads to the objects scattering in the image, not

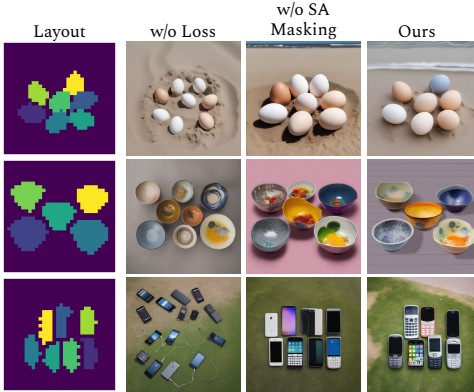

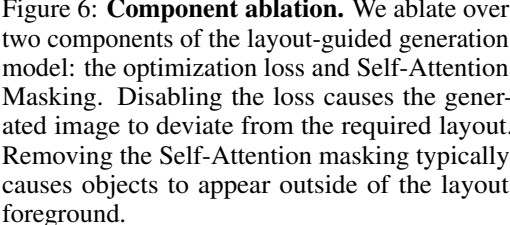

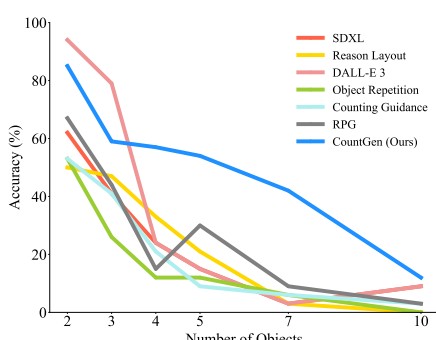

Figure 6: **Component ablation.** We ablate over two components of the layout-guided generation model: the optimization loss and Self-Attention Masking. Disabling the loss causes the generated image to deviate from the required layout. Removing the Self-Attention masking typically causes objects to appear outside of the layout foreground.

Figure 7: **Accuracy, as a function of the number of generated objects.** Accuracy evaluated by human raters, over the set of 200 evaluation images. CountGen (blue) outperforms all methods for $n > 3$, and is on par with DALL-E 3 for 2 and 3 objects.

constrained by the required mask. When removing the self-attention masking the objects tend to obey the mask unwanted object instances occur in the background.

We confirm these observations quantitatively in Table 3, where we evaluate the adherence of the generated image to the input mask. We use YOLOv9 to detect the bounding boxes of generated objects and compare them to the input mask using three metrics: *Precision* is the percentage of bounding boxes that highly overlap (IOU>0.6) the mask (union of all object masks), *Recall* is the percentage of mask pixels that are covered by bounding boxes, and *IOU* is measured between the boxes and the mask. Our findings align with the qualitative observation: removing the self-attention masking leads to a worse precision score, meaning objects are generated in the background. Removing the layout loss leads to low recall and IOU, meaning poor adherence to the mask. CountGen-Image, employing both components, achieves balanced results by generating objects in accordance with the mask. Overall, these results emphasize the critical roles that both components in ensuring accurate adherence to the input mask.

**Pipeline analysis.** We identified three primary sources of failure within our pipeline, as described in Table 6: (1) **Instance Localization**—The number of clusters identified by DBSCAN is incorrect, differing from what is generated if the full denoising process is performed; (2) **CountGen**—The number of instances in its output is incorrect compared to the target number; (3) **Layout Guidance**—The guidance does not produce the target count.

Notably, the CountGen module consistently adds an extra object mask in every case, suggesting that the error are related to either clustering or layout guidance. Out of all the failures, 47 were due to Instance localization and 49 were due to loss. Over-generation occurred mostly for target count $k$ bigger than 5, whereas layout-guidance issues are more frequent with target counts $\leq 5$. Among

Table 2: Model components Accuracy (%).

| Text → Layout | Layout → Image | CoCoCount | | Compbench |
|---|---|---|---|---|
| | | YOLOv9 Acc. | Human Acc. | Human Acc. |
| CountGen | CountGen | **50** | **52** | **48** |
| CountGen | B-Attn | 40 | 42 | 40 |
| Random | CountGen | 37 | 44 | 42 |
| Random | B-Attn | 29 | 30 | 35 |

Table 3: CountGen-Layout components. Error bars represent standard error across 200 images.

| Method | Precision | Recall | IOU |
|---|---|---|---|
| CountGen | **59** ±3.1 | **82** ±2.5 | **52** ±1.2 |
| - SA masking | 48 ±3.1 | 81 ±2.7 | 51 ±1.5 |
| - Layout loss | 49 ±2.9 | 64 ±2.5 | 36 ±1.4 |

the Instance localization failures, we observed that 31% of the errors occurred when more than 15 instances were generated in the original image.

**Sensitivity analysis of Instance-level Features.** To quantitatively evaluate the performance of our instance-localization step we compare the bounding box predictions extracted from our method's instance localization masks to ground truth bounding boxes. We manually annotate these instance-level bounding boxes on a subset of 85 images taken from the CoCoCount dataset. We report standard precision and recall metrics over a range of timesteps (Table 4) and layers (Table 5). The time and layer that we selected on set-aside data, generalize well to the test data, and these hyper-parameters out-perform other choices.

Table 4: Precision and Recall across different timestamps.

| Metric | t=900 | t=800 | t=600 | t=500 (Ours) | t=400 | t=200 | t=0 |
|---|---|---|---|---|---|---|---|
| Precision | 0.81 | 0.88 | 0.88 | **0.92** | 0.90 | 0.90 | 0.83 |
| Recall | 0.51 | 0.79 | 0.84 | **0.92** | 0.93 | 0.93 | 0.89 |

Table 5: Precision and Recall across different layers.

| Metric | down_10 | down_40 | mid_120 | mid_136 | up_48 | up_52 (Ours) | up_70 | up_100 |
|---|---|---|---|---|---|---|---|---|
| Precision | 0.27 | 0.27 | 0.26 | 0.39 | 0.39 | **0.92** | 0.67 | 0.45 |
| Recall | 0.56 | 0.56 | 0.10 | 0.16 | 0.15 | **0.92** | 0.67 | 0.35 |

## 7 LIMITATIONS

Occasionally, our optimization (Section 3.3) results in multiple instances of an object in an area intended for just one by the layout. In other cases CountGen generates plain backgrounds compared to SDXL (Figure 8). In addition, the scope of our experiments may seem narrow, since we focus on generating scenes with up to 10 instances and a single object per prompt. Nevertheless, we have shown in Section 5 that even this setup is highly challenging to contemporary models, especially as the number of instances required to generate grows, as evident by the massive drop in performance, even for DALL-E 3 (see Figure 7).

## 8 CONCLUSIONS

The task of generating images that depict the number of requested objects correctly is a hard task. It requires models to capture "objectness", and obey global spatial constraints, at the same time they generate a well-formed natural image. Current text-to-image diffusion models perform poorly in this task (Table 1), especially when asked to generate more than three objects (Figure 7).

Our CountGen approach took three steps to address this task. First, we identified a notion of objectness from the self-attention layers of the diffusion model. Then, we trained a U-Net model that learned to correct the number of instances of an object in a given layout, whether it is removing or adding instances of an object such that the structure of the layout is preserved. Third, we developed a layout-guidance optimization method method to generate images from the corrected layout.

Together, this approach almost doubled the counting accuracy from 26% in standard SDXL to 52% using our method applied to SDXL. We expect the lessons learned from this method, specifically the features that represent objectness and the process of learning to automatically fix a layout, to become useful in other problems of structured generation like spatial constraints in text-to-image models or spatio-temporal constraints in video generation.

## 9 ETHICS STATEMENT

For crowdsourcing experiments and research with human subjects, the paper includes the full text of instructions given to participants and screenshots. Our protocols are described in the main paper and screenshots of the experiments and questions for raters are included in our supplemental material. The following qualifications were used to choose annotators

- HIT Approval Rate (%) for all Requesters' HITs is greater than equal to 99.
- Number of HITs Approved is greather than 5000.
- Annotator successfully passed a qualification test.

## 10 REPRODUCIBILITY STATEMENT

Our method, CountGen, is thoroughly described in Section 3. Compute details and hyperparameters are provided in Appendices A and B. Detailed definitions for training the U-Net are provided in Appendix C.2, and evaluation of ReLayout is in Appendix C.3.

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

# A  APPENDIX

**Efficiency.**    CountGen takes ~36 seconds on average to generate an image on a single A100 80GB. We arrive at this number by iterating over CoCoCount. To put in context, Bounded-Attention (Dahary et al., 2024a) takes ~55 seconds and requires bounding boxes as input, while our solution is not input-dependent. SDXL takes ~8 seconds.

**Compute.**    All experiments were conducted over a period of a week on a single A100 80GB.

# B  IMPLEMENTATION DETAILS & REPRODUCIBILITY

## B.1  COUNT NUMBER EXTRACTION

To accurately extract count numbers from the textual prompts, we employ spaCy's dependency graph parser (Honnibal & Montani, 2017) to identify and isolate indices of related subjects and numeric modifiers. This methodology is inspired by the approach detailed in "Linguistic Binding in Diffusion Models" by Rassin et al. (2024), which demonstrates the automated extraction of subjects and their attribute modifiers. We have adapted this technique to specifically recognize numeric modifiers, both spelled out (e.g., "five dogs") and in numeral form (e.g., "5 dogs"). This adaptation ensures that each numeric modifier is correctly associated with its corresponding noun, thereby facilitating accurate cross attention in our model's processing pipeline.

## B.2  COUNTGEN

**Layout guided generation.**    In our implementation, the self-attention masking is applied at timesteps $t \in [1000, 900]$, in the decoder layers of the U-Net. The object layout loss is applied at timesteps $t \in [1000, 500]$, in all layers of the U-Net. Our pipeline used the Attend-and-Excite (Chefer et al., 2023) code base as a starting point.

**ReLayout.**    The ReLayout U-Net was built upon the U-Net Implementation of (Buda et al., 2019). We trained the U-Net with a learning-rate of *8e-6*, a batch-size of size *32* and the Adam optimizer. The intersection penalty is set to *0.25* and the Dice penalty is set to *1*. During training we apply a horizontal flip augmentation across all masks, and shuffle augmentation where we randomly re-arrange the input channels.

**Instance identification.**    In the DBSCAN clustering algorithm, we used a dynamic epsilon value in the range of $[0.1, 0.2]$ and used cosine similarity as the distance metric.

## B.3  COMPARED METHODS

Each prompt in CoCoCount and T2I-CompBench-Count was assigned a unique random seed and was used by all baselines and CountGen.

We compared CountGen with the following baselines:

**SDXL (Podell et al., 2023).** We used the `stable-diffusion-xl-base-1.0` model.

**Repeated Object.** In this baseline, we used the same model and seeds as in SDXL but modified the prompts. We repeated the object in the prompt as many times as the target count. For example, "a photo of three cats" was changed to "a photo of a cat and a cat and a cat".

**Reason Out Your Layout (Chen et al., 2023).** This baseline has two main steps. First, it leverages `GPT-3.5-turbo` to generate spatially reasonable coordinates to be used as a bounding box for each instance of an object (i.e., "a photo of three cats" results in three bounding boxes, one for each cat). Second, it uses the generated layout to guide the generation process. We followed the prompt used by the authors, however, it seems that the responses by `GPT-3.5-turbo` and the author's parser are not completely cohesive, which at times leads to zero bounding boxes. We count such cases as failures. For the CoCoCount experiment, it successfully generated 134/200 images, and for

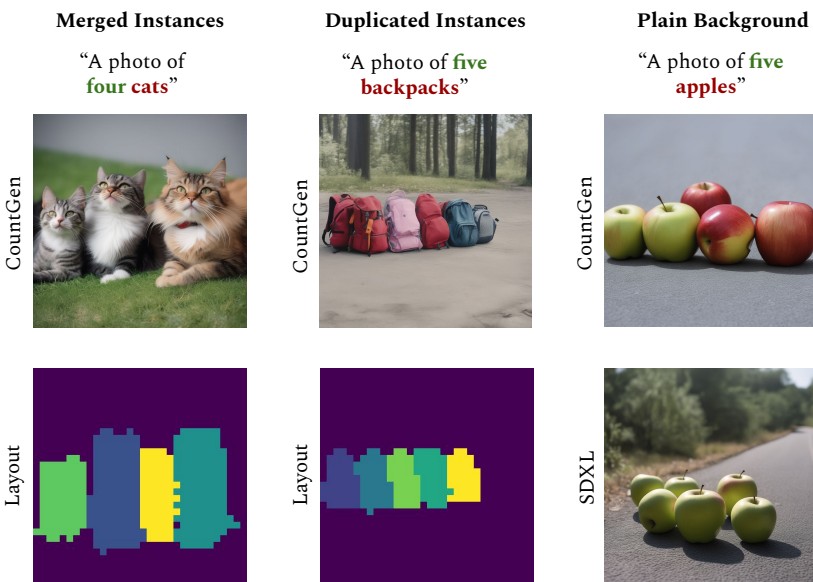

Figure 8: **Limitations.** Failure modes of CountGen.

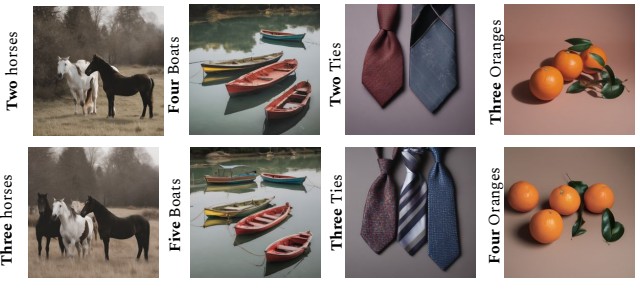

Figure 9: **A training set for a Re-Layout.** We created pairs of images using SDXL, using the same seed and prompts that only differ by object count. We filtered out images that did not conform to the prompt, using the techniques described in Section 3.1. The resulting image pairs preserve the scene and layout except adding one object.

T2I-CompBench-Count, just 89/200. Failures were counted as errors in the reported results. We did not need to make changes to the code to run it.

**DALL-E 3 (Betker et al., 2023).** We used the OpenAI API interface for the DALL-E 3 model with "standard" image quality. We did not use seeds in this baseline.

**Random masks + BoundedAttn (Dahary et al., 2024a).** Given a prompt with a required number of object instances, we create a corresponding layout with the correct number of objects randomly placed in the image plane in a way they do not intersect one another. Then we used Bounded Attention to generate an image condinitioned on that layout.

**Counting Guidance (Kang et al., 2023).** The authors provided us with their code. We did not need to change it to run our experiments.

**RPG (Yang et al., 2024).** We used the official code, with SDXL and GPT-4 for our experiments.

## C EXTENDED DETAILS ON COUNTGEN

### C.1 RELAYOUT: MATCHING OBJECTS

We aim to understand how $M_k^i$ transitions to $M_{k+1}^i$. Specifically, for each object $i \in 1, \ldots, k$ in the original $M_k$ layout, our ReLayout objective is designed to predict how the corresponding mask $M_k^i$ changes in the new image $M_{k+1}^i$, and additionally where to insert the added object $k+1$. This design encourages the model to slightly modify existing objects while preserving spatial and shape consistency across the images.

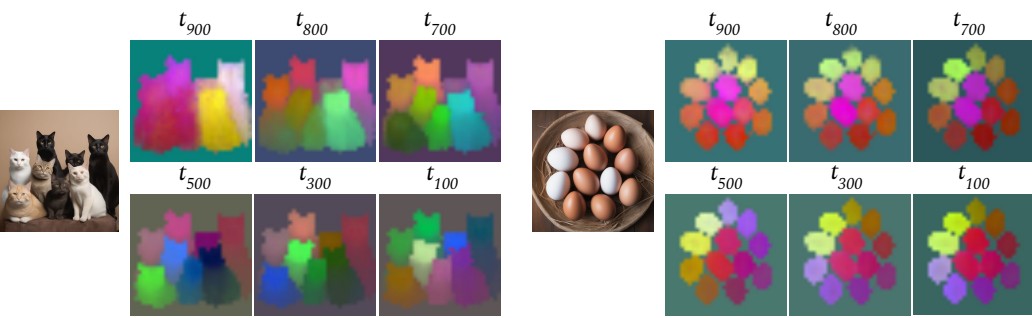

Figure 10: **PCA visualization across timestamps** to explore the notion of objectness inside SDXL latent space, we visualize a dimension-reduced self-attention feature maps across different timestamps range from $t = 900$ to $t = 100$. Initially, up to timestamp $t = 500$, clear separation is not observed in some objects (e.g., some eggs appear in similar colors). However, starting from $t = 500$, a distinct separation emerges, with each object clearly distinguished by different shades.

To this end, we first have to establish a correspondence between the object masks $(M_k, M_{k+1})$. We employ the Hungarian algorithm (Kuhn, 1955) to find the optimal one-to-one matching between masks in the two images based on the overlap and similarity of the masks. This algorithm effectively pairs each object in $M_k$ with a corresponding object in $M_{k+1}$. The object in $M_{k+1}$ that remains unmatched represents the additional object introduced in the new image, providing a clear identifier for the increment in object count.

## C.2 Losses for Training the ReLayout

We use two training losses:

*Dice Loss:* measures the overlap between the predicted mask and target mask across all channels containing objects:

$$L_{\text{Dice}}^i = 1 - \frac{2 \sum_{p \in P} M_{k+1}^i(p) \cdot M_{k+1}^{*i}(p)}{\sum_{p \in P}(M_{k+1}^i(p) + M_{k+1}^{*i}(p))} \tag{2}$$

Here, $p$ iterates over all pixels $P$ in the masks, and $i$ ranges over all possible object channels. For all $k + 1$ channels, the total dice loss is:

$$L_{\text{Dice}} = \sum_{i=1}^{k+1} L_{\text{Dice}}^i \tag{3}$$

*Intersection Loss:* To ensure distinctiveness among the predicted masks and to minimize overlap between different object masks, the intersection loss for all possible pairs of different masks in the output mask containing objects is given by:

$$L_{\text{Overlap}} = \sum_{i=1}^{k+1} \sum_{j \neq i}^{k+1} \frac{2 \sum_{p \in P} M_{k+1}^i(p) \cdot M_{k+1}^j(p)}{\sum_{p \in P}(M_{k+1}^i(p) + M_{k+1}^j(p))} \tag{4}$$

## C.3 ReLayout Evaluation

We use two metrics for the evaluation:

**Extra mask median score.** To calculate the extra mask size score, we first find the median size $(S_{\text{median}})$ of all object masks. We then compare this to the size of the new mask $(S_{\text{extra}})$. The score is defined as:

$$\text{Score} = \frac{\min(S_{\text{extra}}, S_{\text{median}})}{\max(S_{\text{extra}}, S_{\text{median}})}$$

which gives a value between 0 and 1. A score closer to 1 indicates that the new object's size is more similar to the median-sized object. For ReLayout, the score is 0.705, indicating that the new object has become more similar in size to the other objects in the scene.

**Average intersection score.** This metric measures the average intersection between an object $i$ and all other object masks $j$, normalized by the size of object $i$. A lower score indicates less overlap between objects. During training, this score decreased to 0.18, indicating small intersection between the objects.

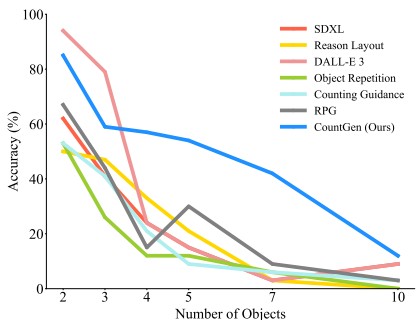

Figure 11: **Accuracy, as a function of the number of generated objects.** Accuracy evaluated by human raters, over the set of 200 evaluation images. CountGen (blue) outperforms all methods for $n > 3$, and is on par with DALL-E 3 for 2 and 3 objects.

Figure 12: **Accuracy, as a function of the number of generated objects.** Accuracy evaluated by YOLOv9, over the set of 200 evaluation images. Here, CountGen (blue) outperforms all methods.

Table 6: Failure Analysis across Different Target Counts

| Target Count | Instance Localization Failures | Loss Failures | Total Failures |
|---|---|---|---|
| 2 | 2 | 3 | 5 |
| 3 | 4 | 10 | 14 |
| 4 | 5 | 9 | 14 |
| 5 | 8 | 7 | 15 |
| 7 | 11 | 8 | 19 |
| 10 | 17 | 12 | 29 |

### C.4 DATASETS

**CoCoCount.** To create this set, we first select at random 20 classes from MSCOCO (Lin et al., 2014). We then sample from six counting categories: 2,3,4,5,7, and 10. The two and three categories contain 34 samples, while the rest contain 33. Our prompts consist of the pattern "a photo of {number} {object}" with an optional variation of scenes: "on the grass", "on the road", or "on the ground", which we incorporate for 50% of the prompts, also randomly. In total, we have 200 prompts. Below are the complete lists from which elements were chosen:

**Objects:** 'car', 'airplane', 'bird', 'cat', 'dog', 'horse', 'sheep', 'cow', 'elephant', 'bear', 'backpack', 'tie', 'sports ball', 'baseball glove', 'cup', 'bowl', 'apple', 'donut', 'cell phone', 'clock'. **Counting Categories:** 'two', 'three', 'four', 'five', 'seven', 'ten'. **Scenes:** 'on the grass', 'on the road', 'on the ground'.

## D EVALUATION

**Automatic evaluation.** We use the implementation by Ultralytics YOLO of YOLOv9e (large).

### D.1 HUMAN EVALUATION

We use the Amazon Mechanical Turk platform and ensure the evaluation is of high quality by hiring raters with a minimum of 5,000 approved HITs and an approval rate exceeding 98%. Each example was shown to three raters and the majority selection was taken. The compensation was $15 per hour. Screenshots of the count precision task can be viewed in Figure 13, Figure 14, Figure 15 and the image fidelity task in Figure 16.

**Instructions for the Image Evaluation Task**

Welcome to the Image Evaluation Task! Your role is crucial in counting the number of objects in an image. Please follow the instructions carefully:

1. **Examine the Image**: You'll be presented with an image. Pay close attention to the objects and details within the image, irrespective of its overall appearance. **Remember, an image can look unusual or "weird" but still accurately answer the questions.**

Figure 13: Instructions for the Image Evaluation Task - Part 1.

2. **Read the Questions**: Below the image, you'll find a set of questions.
   ○ Are there books in the image?
   ○ If yes, are the books well-formed?
   ○ How many books are there?

**Understanding the Concepts**

For clarity, let's break down some of the key terms:

- **Object in the Image**: This pertains to the visible items or subjects in the provided image. Using the previous example, you'd need to determine if you can see at least one book in the image.
- **Well-formed Objects**: Here you should identify if the focused object is well-formed and can be counted. For instance, in the left example above, the books are well-formed and can thus be counted, but some of the books in the right example are not well-formed. As a result, and it is not clear how many books are there in total.
- **Count**: The count describes the quantity of an object. For instance, in example (1) above, there are 6 books.

Please ensure accuracy in your evaluation. Inaccurate or rushed answers may be rejected.

Thank you for your attention to detail and dedication to this task!

Figure 14: Instructions for the Image Evaluation Task - Part 2.

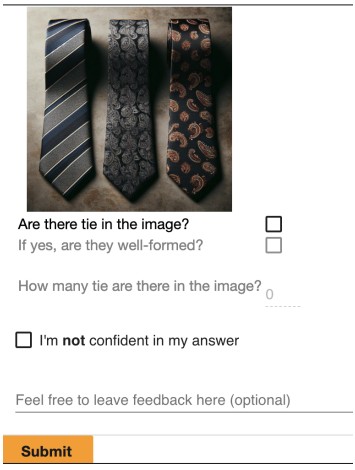

Are there tie in the image? ☐
If yes, are they well-formed? ☐

How many tie are there in the image? 0

☐ I'm **not** confident in my answer

Feel free to leave feedback here (optional)

**Submit**

Figure 15: Example task to count the number of objects and assess their well-formedness.

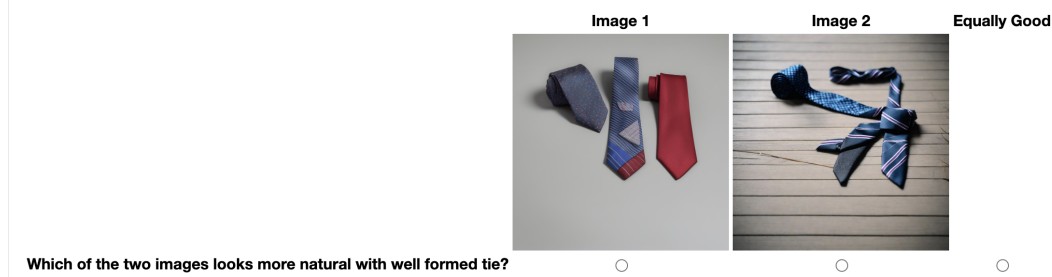

**Which of the two images looks more natural with well formed tie?**

Figure 16: Example task to compare image fidelity based on prompt matching and naturalness.

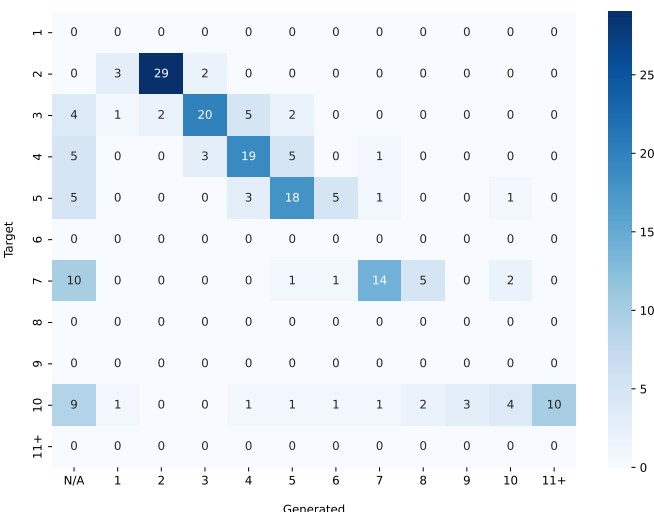

Figure 17: Confusion matrix from human evaluation (Section 4) of the count accuracy experiment for CountGen. As noted in Figure 15, evaluators could indicate if they were unsure of their response ("N/A" in the table).

