# OpenReview forum: "Make It Count: Text-to-Image Generation with an Accurate Number of Objects"
_ICLR.cc/2025/Conference — ICLR 2025 Conference Withdrawn Submission_

### Official Review · Reviewer_CXh8 · 2024-10-19

**Soundness:** 2
**Presentation:** 3
**Contribution:** 2
**Rating:** 3
**Confidence:** 4

**Summary:**

This paper addresses the problem of generating a precise number of objects in Diffusion models.  From my understanding, it introduces: 1) an instance segmentation framework based on Attention; 2) a network for modifying the layout (especially in cases where the count of objects needs to be increased); and 3) a well-designed guidance method to direct the image generation towards the modified layout. The authors conducted main experiments on two benchmarks (one of which was introduced in this paper) as well as ablation studies, demonstrating the effectiveness of the proposed method.

**Strengths:**

This article focuses on a subproblem within diffusion inference process——how to generate images with a more precise number of objects. The authors provide a detailed discussion of related work, including Counting Guidance [1] and RPG [2]. They also introduce a new benchmark, which contributes to advancing research within the whole community. Lastly, the paper covers a wide range of techniques, including improving guidance, enhancing instance segmentation, and assembling these components into a pipeline. Particularly, their discussion on "Self-attention masking" is interesting.

[1] Kang, Wonjun, Kevin Galim, and Hyung Il Koo. "Counting guidance for high fidelity text-to-image synthesis." arXiv preprint arXiv:2306.17567 (2023).

[2] Yang, Ling, et al. "Mastering text-to-image diffusion: Recaptioning, planning, and generating with multimodal llms." Forty-first International Conference on Machine Learning. 2024.

**Weaknesses:**

1.The comparison of baselines is unfair.

In Table 1, the authors compare CountGen with other baselines, but CountGen is based on the SDXL model, while some of these baselines are not. For example, the base model for Reason Out Your Layout [1] is SD1.4. And Counting Guidance [2], as a further improvement of Universal Guidance, uses SD1.5 or SD2. If you haven’t modified its source code (as mentioned in Line 797), it means the base models are different, making this comparison unfair. Therefore, migrating CountGen to other SD models for a fair comparison should be part of your work. Otherwise, this kind of unfair comparison is detrimental to the progress of the entire field. I think you should conduct experiments on other Diffusion Base Models (e.g, sd1.4, sd1.5, sd2) to achieve fair comparisons. It's worth noting that this is not an unnecessary increase in experimental workload.

2.No quantitative experiments in T2IBenchmark.

The authors use both human metrics and YOLO metrics. They apply these metrics in the CoCoCount benchmark, but what confuses me is why they do not use objective quantitative indicators (YOLO) in the T2ICompBench, and only report human evaluation scores in the paper. Given that CoCoCount is one of the contributions of this paper and has not yet been publicly released, the quantitative evaluation only on the self-built dataset is not convincing. I hope the authors can first explain why they only report human evaluation results in T2IBench? What is the reason for this? If possible, further provide the evaluation results of T2IBench.

3.The importance of the "Corrected Layout" is not sufficiently emphasized.

The core of CountGen lies in first providing the original layout and then correcting it. What is the advantage of CountGen over models like RPG[3]?  CountGen first detects the incorrect layout and then corrects it, finally guiding the generation. Why not directly plan the layout from the beginning and then directly guide the generation using this layout?  I guess the author's idea is similar to the work in OMG [4], where they first use the original capabilities of Diffusion to provide a layout and then modify it on this basis, possibly improving the quality. However, this paper does not provide sufficient quantitative analysis to compare the advantages and disadvantages of the "GPT-Based-Layout" and "Corrected Layout" methods, but only conducts ablation experiments on "Random Layout".  I think this paper should further explain the necessity of using the "Corrected Layout" and provide qualitative or quantitative ablation experiments for comparison.

Additionally, the "ReLayout" module can only handle the case where the number of extra objects is "1" (this is due to the setup of the training dataset, see line 266). However, in Diffusion generation, there are often more than one extra objects, for example, generating 2 cats but the actual generated image has 4 cats. In such cases, ReLayout cannot be applied, so the universality of this method is not strong. The authors might try to construct more diverse datasets and add conditional quantity signals in ReLayout to output the corrected Layout.

4.The selection of UNet features lacks generality.

During the first stage of object segmentation, the authors use a specific layer in UNet and empirically believe that this layer works the best. Similarly, like weak 1, is this layer the best for sd, sd2, and sdxl? If not, then for each different Diffusion Model, the optimal layer would have to be manually and empirically selected again, increasing the difficulty of deploying this method. I think the authors should further research this part to enhance the generality of feature selection.


[1] Chen, Xiaohui, et al. "Reason out your layout: Evoking the layout master from large language models for text-to-image synthesis." arXiv preprint arXiv:2311.17126 (2023).

[2] Kang, Wonjun, Kevin Galim, and Hyung Il Koo. "Counting guidance for high fidelity text-to-image synthesis." arXiv preprint arXiv:2306.17567 (2023).

[3] Yang, Ling, et al. "Mastering text-to-image diffusion: Recaptioning, planning, and generating with multimodal llms." Forty-first International Conference on Machine Learning. 2024.

[4] Kong, Zhe, et al. "OMG: Occlusion-friendly Personalized Multi-concept Generation in Diffusion Models." arXiv e-prints (2024): arXiv-2403.

**Questions:**

1.I hope the authors can achieve fair comparisons by comparing different baselines under the same base model (e.g., sd1.5 or sd2). For example, you can try to migrate CountGen to sd1.5, or migrate the baselines to be compared to sdxl.

2.I hope the authors can explain why they did not include experiments with the YOLO metric in T2IBench. And I hope they can include this experiment in the new version.

3.I hope the authors can explain the advantages and disadvantages of ReLayout and GPT-Based Layout, and clarify the novelty of CountGen.

4.I hope the authors can explain the reason for selecting l_{52}^{up} in UNet and whether it has generality.

If these confusions are clarified, I will consider raising the score rating.

**Details Of Ethics Concerns:**

This paper does not have any ethical concerns that are worrisome. The dataset it proposes is based on the CoCo Datasets, so it not only has the risk of privacy leakage.

---

### Official Review · Reviewer_kbLb · 2024-10-29

**Soundness:** 2
**Presentation:** 3
**Contribution:** 1
**Rating:** 3
**Confidence:** 4

**Summary:**

This paper presents a pipeline designed to facilitate counting tasks using pre-trained text-to-image diffusion models. The pipeline consists of three stages: instance localization, ReLayout, and layout-guided image generation. The results demonstrate that the proposed approach significantly outperforms all state-of-the-art baselines in counting tasks.

**Strengths:**

The problem that this paper is addressing is critical, as all SOTA image generative models fail in solving counting tasks. The authors attempt to solve this problem by adopting a pre-trained text-to-image diffusion model during inference time to correct the number of objects expected in the scene. Hence, no costly pretraining is needed for unlocking these new abilities. The paper addresses the task of generating images using a specific prompt format, such as "A photo of [number of objects] [object type]." It conducts numerous evaluations and ablation studies to demonstrate the effectiveness of each component. In my view, all these parts are highly effective in enabling the diffusion model to accurately count objects.

**Weaknesses:**

I think the paper introduces many exciting components that address the counting problem. However, I believe the paper suffer from a serious shortcoming which should be discussed/addressed in the next version of the paper:

**Overfitting to one specific prompt format**: In all visualizations and explanations of the paper, I notice that it consistently uses a prompt template structured as "A photo of [number of objects] [object name]." For example, "A photo of five apples" or "A photo of five backpacks." Why only use this prompt template in all ablations and qualitative visualization? What about exploring additional prompt templates, such as those provided in the T2I-CompBench dataset? I would suggest authors to discuss about following points during rebuttal or future submissions:

1. Explicitly state the limitations of their current approach in handling more complex prompts.
2. Provide results on a wider range of prompt templates from T2I-CompBench, even if performance is lower.
3. Discuss potential ways their method could be extended to handle multiple object types and attributes in future work.

More specifically, given following prompts:

```
Prompt 1: one apple and two oranges
Prompt 2: four balloons, two rabbits, three cars, four butterflies, and a dog created a festive atmosphere
Prompt 3: three printers, four people, a bicycle, three microwaves, and two refrigerators cluttered the office
```

Can authors:
1. show results of their instance localization, ReLayout, and layout-guided generation steps on 2-3 example prompts with multiple object types
2. discuss which components of their pipeline would need to be modified to handle these more complex prompts
3. quantify performance on a subset of T2I-CompBench prompts with multiple object types, if possible

***Note***: The above issue is concerning because I believe this solution cannot scale to other prompt templates, which require more object types, each associated with an attribute, such as "A photo of 6 green apples." If the authors believe their proposed method can effectively generate different objects from various categories within a scene, I would ask them to provide this information during the rebuttal. Otherwise, I believe this paper has limited contributions/applications.

**Questions:**

Asked above.

---

### Official Review · Reviewer_dTMu · 2024-11-01

**Soundness:** 2
**Presentation:** 3
**Contribution:** 2
**Rating:** 6
**Confidence:** 3

**Summary:**

This paper presents a novel method, CountGen, which is a two-stage method for improving the accuracy of object counts in text-to-image generation using diffusion models.
The first stage identifies an “objectness” representation in the SDXL model by analyzing self-attention features at layer $L_{52}^{up}$ at time t=500. It then uses cross-attention maps to locate and cluster object instances, creating an initial layout.
The second stage, ReLayout, corrects this layout for over-generation or under-generation of objects.
For under-generation, ReLayout employs a U-Net trained on layout pairs with varying object counts, adding objects while preserving scene composition.
Finally, layout guidance, using object layout loss and self-attention masking, directs the generation process to adhere to the corrected layout. This approach significantly improves count accuracy, especially for larger numbers of objects, without compromising image quality.

**Strengths:**

1. The authors identify a specific layer $L_{52}^{up}$ within the SDXL model that exhibits strong object instance separability in its self-attention features.
This discovery is significant because it reveals an inherent capability of diffusion models to represent individual objects as distinct entities, which is crucial for accurate counting.
The paper provides evidence for this through PCA visualization of self-attention feature maps.

2. The paper proposes a novel ReLayout module to correct the number of object instances in the generated layout. This module tackles both over-generation and under-generation. This paper also introduces a layout-guided generation process that combines object layout loss and self-attention masking. The object layout loss encourages object creation within the defined masks, while self-attention masking prevents objects from being generated in the background. All these techniques ensures that the final image closely adheres to the corrected layout.

3. The paper introduces CoCoCount, a new dataset designed specifically for automatic evaluation of count accuracy. Based on this new benchmark, it presents a comprehensive evaluation of CountGen on the CoCoCount and T2I-Compbench-Count benchmarks.

**Weaknesses:**

1. CountGen relies on the specific architecture of the SDXL model, particularly the self-attention features extracted from layer $L_{52}^{up}$. This dependence raises concerns about the method's transferability to other diffusion models. Further investigation is needed to assess whether the concept of objectness, as captured by CountGen, can be generalized to different architectures or whether model-specific adaptations would be required. Since the PCA visualization in Fig.3 should be easily reproduced using other models, it could be verified in time.

2. The ReLayout module is trained on a dataset of layout pairs generated by SDXL using the same random seed but slightly different object counts. This approach, while clever in its utilization of consistent layout generation by SDXL, could potentially lead to overfitting. ReLayout might learn to exploit specific artifacts or biases in SDXL's output, limiting its effectiveness when applied to layouts generated by other models or even by SDXL with different seeds. For example, how to make sure that SDXL is able to always generate correctly with "two apples" and "three apples" with the similar layout. Over here, I'm not sure if any VLM model (like visual question answering) could help.

3. While the paper briefly mentions the time taken by CountGen to generate an image (~36 seconds), it lacks a detailed discussion of the computational cost compared to other methods, especially in relation to the achieved accuracy gains.

**Questions:**

Please refer to the weakness.

I would like to say this paper is solving a good problem existing in T2I diffusion models. However, this method needs to create a dataset and train the ReLayout module for each kind of T2I diffusion model, which limits its application scenario. As a comparison, the paper “Counting Guidance for High Fidelity Text-to-Image Synthesis” doesn't need the training process.

---

### Official Review · Reviewer_cpyo · 2024-11-04

**Soundness:** 2
**Presentation:** 2
**Contribution:** 2
**Rating:** 3
**Confidence:** 4

**Summary:**

Existing text-to-image models struggle with generating a certain number of objects in images. The paper attributes this inability to keep a sense of separate identity for every instance of the object. The paper resolves this problem by first identifying features within a diffusion model, SDXL, that carry object identity information. Then the paper proposes CountGen, which counts instances of objects during denoising, detects over- and under-generation, and fixes the latter by predicting missing objects to guide denoising in order to obtain a correct object count. Results on two benchmarks show that CountGen outperforms existing baselines in the accuracy of generating a specified number of objects.

**Strengths:**

* The work identifies features that represent object instances rather than semantics in SDXL, offering a new understanding of the internal representations of diffusion models.
* The work proposes CountGen, a technique to guide SDXL for accurate number of object instances, with a novel object layout loss and self-attention masking strategy.
* Detailed benchmarking demonstrates that CountGen improves accuracy in counting without degrading image quality.

**Weaknesses:**

* The analysis in the introduction is unclear. Speficically, the work identifies the cause of inability to generate a specified number of objects to be the lack of instance-level features (L61-65). However, the work later discovers the presence of such features (L199-210), which contradicts the claim. The fact that previous works have not found such features is not an explanation on why current diffusion models do not perform well in counting tasks.
* Lack of comparisons with existing baselines. The ReLayout proposed in this work is very similar to the closed-loop generation proposed in Self-correcting Diffusion Models [1], which aims at solving problems in a larger scope that includes count-specific generation. The authors are recommended to discuss the relevance and compare with [1] in terms of performance.
* The proposed method requires finding features that are related to instanceness for each model. Furthermore, the authors did not provide explanations on the process of finding such features, which are the core of the method.
* The authors only experimented with SDXL, which does not show generalization capabilities of the proposed method.
* The proposed method introduces an additional module to generate the layouts. This might increase the run-time, impacting downstream use cases. The authors are recommended to show the wall-clock runtime and compare with previous baselines.

[1] Self-correcting LLM-controlled Diffusion Models. T. Wu, et al. CVPR 2024. https://arxiv.org/abs/2311.16090.

**Questions:**

* Does CountGen increase inference time compared to baselines? If so, how does its runtime compare to the one of previous methods?
* How does CountGen compare with related works in terms of performance mentioned in the section above?
* Is the discovery that instance feature exists in text-to-image diffusion models generalizable (i.e., not just for SDXL)?

---

### Note · Authors · 2024-11-14

**Comment:**

We would like to thank the reviewers for their valuable feedback on our manuscript.

We are pleased that the reviewers recognized the significance of our new insights into the internal representations of diffusion models, specifically highlighting their capacity to represent individual object instances. Additionally, we appreciate that the reviewers found our proposed components, such as the ReLayout module, to be novel. Finally, we thank the reviewers for acknowledging the importance of the task and the overall success of our method in improving current benchmarks.

While we have decided to withdraw our work, we would like to address some of the points raised by the reviewers:


### 1. **Instance-level Feature Analysis**
Instance-level feature analysis is a core contribution of our paper. In Section 6, we provide a quantitative analysis across layers and timesteps, with qualitative demonstrations in Figures 3 and 10. In a future version of our paper, we plan to expand this discussion to better illustrate the methodology for identifying such representations in SDXL and in other models. It is important to clarify that our findings do not imply that diffusion models are incapable of solving the accurate-counting problem due to a lack of instance-level features. Rather, our results demonstrate that these representations do exist within the models, which is a novel and non trivial finding. However, explicit interventions may be necessary to enable the models to fully leverage these features.


### 2. **Generalization to other models**
While it is very common for studies in the text-to-image field to construct methods on top of  a specific base model, we recognize the importance of exploring whether the internal representations we identified in our paper also emerge in other diffusion models. We intend to investigate this question in a future version of our paper.


### 3. **The ReLayout Module**
We are pleased that the reviewers found our ReLayout module to be novel. It is important to note that the ReLayout module is agnostic to prompt templates, as it operates solely on object layouts. Although we constructed the data using SDXL, we observed that the model generalizes well to unseen classes and templates. In Section 3.2, we provide a detailed description of the dataset construction process required to train the module and explain how we ensure that only input and output layouts that align are selected.


### 4. **Computational Cost**
We provide a report on computation time in Appendix A. In a future version of our paper, we plan to include a more detailed comparison of computational costs relative to other methods.


### 5. **T2IBench Evaluation**
In our paper, we provide both human and YOLO-based evaluations. However, for the T2IBench dataset, we only conducted a human evaluation. T2IBench cannot be evaluated with a YOLO-based approach because it includes classes beyond those in the COCO dataset, unlike our proposed CoCoCount benchmark. In other words, the YOLO-based detection model, which is trained on COCO classes, cannot fully operate on T2IBench. Additionally, our experiments show that existing open-vocabulary detectors are not reliable enough for counting-based evaluations. This limitation motivated us to create the CoCoCount benchmark, which allowed for an automatic evaluation using the COCO trained YOLO model. Therefore, we opted to evaluate T2IBench using human evaluations, which tend to be more precise.

**Withdrawal Confirmation:**

I have read and agree with the venue's withdrawal policy on behalf of myself and my co-authors.